# Sequential Defense Strategies: From Ant Recruitment to Leaf Toughness

**DOI:** 10.3390/plants14010049

**Published:** 2024-12-27

**Authors:** Danilo F. B. dos Santos, Eduardo S. Calixto, Helena M. Torezan-Silingardi, Kleber Del-Claro

**Affiliations:** 1Intercollege Graduate Degree Program in Ecology, The Huck Institutes of the Life Sciences, The Pennsylvania State University, University Park, PA 16802, USA; santosdanilo499@gmail.com; 2Entomology and Nematology Department, University of Florida, Gainesville, FL 32608, USA; calixtos.edu@gmail.com; 3Programa de Pós-Graduação em Ecologia e Conservação de Recursos Naturais, Instituto de Biologia, Universidade Federal de Uberlândia, Uberlândia 38408-100, MG, Brazil; hmtsilingardi@gmail.com

**Keywords:** defense turnover, herbivory, leaf toughness, extrafloral nectaries

## Abstract

Plants express many types of defenses in response to herbivory damage. These defenses can be displayed as a sequence or they can overlap, increasing efficiency in protection. However, leaf defense shifts during leaf development, including extrafloral nectaries (EFNs), are neglected in natural tropical systems. To address this gap, our study evaluates the shifts in defense strategies of *Eriotheca gracilipes*, focusing on extrafloral nectaries and leaf toughness across leaf development stages. We also simulate herbivory by cutting the leaves to address the role of visiting ants against herbivores. We observed that *E.* g*racilipes* exhibits a defense turnover, shifting from indirect defenses (e.g., EFNs) in young leaves to physical defenses in adult leaves. Simulated herbivory led to heightened ant visitation, which correlated with decreased herbivory rates, indicating that ant recruitment acts as an effective deterrent. We observed a peak of EFN activity in young leaves, increased foliar toughness in adult leaves, and reduced herbivory on ant-patrolled young leaves. Additionally, *E. gracilipes* demonstrated tolerance to up to 10% foliar loss with no significant impact on leaf asymmetry, although 50% foliar loss increased asymmetry in newly flushed leaves. These results highlight *E. gracilipes’* adaptive flexibility by attracting protective ants when vulnerable and enhancing structural defenses as leaves develops, *E. gracilipes* minimizes herbivory impact. This study provides valuable insight into the adaptive roles of EFNs and tolerance in *E. gracilipes*, contributing to a broader understanding of plant defense strategies.

## 1. Introduction

Plant defensive traits have evolved in response to several selective pressures, particularly those exerted by herbivores [1]. In response to herbivory, plants developed mechanisms that impose resistance, such as secondary metabolites and physical structures (e.g., terpenoids and trichomes, respectively) [2,3,4], which are classified as direct defenses [2]. Plants can also employ indirect defenses via attraction or changes in the behavior of predators, parasitoids, and parasites of herbivores (see review in [5]). One of the most widespread indirect defenses is the protective mutualism between plants and ants mediated by extrafloral nectaries (hereafter, EFNs) [6,7]. In this specific interaction, plants offer a carbohydrate-rich liquid produced by the extrafloral nectaries [8] and, in turn, the ants protect the plant against natural enemies [7,8]. In addition to ants, EFNs can also attract other different predatory arthropods, such as spiders [9,10] and wasps [11,12], which can also provide protection against herbivores.

Another strategy used by plants are induced defenses, in which after damage or risk of attack, plants can intensify some defensive traits [13,14,15,16]. For instance, Ref. [14] showed that after simulated herbivory, EFNs located on the flower petiole of the Brazilian tree *Qualea multiflor*a (Vochysiaceae) increased nectar concentration and volume by averages of 2.8 and 6.6, respectively, attracting a greater number of ants than other parts of the plant. Induced defenses are commonly considered a cost-saving strategy but can also show some disadvantages, such as the time between the damage and the expression of the defense, which can leave the plant vulnerable [17,18]. Therefore, plants might not only rely on one type of defense or strategy to avoid the attack of herbivores.

Even with a large arsenal of defensive mechanisms, herbivores are still able to overcome plant defenses [18]. Thus, alternative strategies can be applied by plants, such as tolerance, i.e., the capacity to maintain fitness levels even after attack [19,20,21]. Damage to a plant, such as from herbivory, only becomes critical when it impairs these processes to the point where the plant can no longer maintain adequate growth and resource acquisition. Therefore, if insects can overcome all defensive strategies imposed by plants, plants can still have no effects on their fitness by tolerating some levels of damage.

These different defense strategies can be expressed separately or overlapping with other defenses during plant ontogeny [2,22,23]. For example, Ref. [24] showed a turnover of defenses in *Qualea multiflora* (Vochysiaceae) over leaf development. These authors observed that there is a shift in the intensity of defenses over leaf development. Non-glandular trichomes are highly expressed in young leaves, while EFNs begin to produce extrafloral nectar and attract ants in intermediate leaf stages. Lastly, leaf toughness is at its highest expressions in adult leaves. This shift in defenses provides effective protection against herbivores over leaf development. Although studies have reported changes in the expression of different defenses throughout leaf and plant ontogeny (see [2,24]), our understanding of how the strategies of escaping, inducing, and tolerating damages are expressed throughout leaf ontogeny is poorly known. For instance, how do leaves minimize herbivore attack along their development? If they are attacked, can the expression of some defenses increase resistance or can plants tolerate these damages without any decrease in fitness? How is the effectiveness of different defenses over leaf development and plant ontogeny? To answer these questions, we carried out several manipulative experiments and surveys using a common EFN-bearing Brazilian Cerrado tree, *Eriotheca gracilipes* (Malvaceae). We evaluated if (i) there is a turnover in the defensive mechanisms throughout leaf development; (ii) ants can act as an indirect defense due to EFN presence; (iii) foliar simulated herbivory has a positive impact on ant abundance, consequently decreasing herbivore numbers (evidence for induced defense); and (iv) different intensities of foliar simulated herbivory have different impacts on new flushing leaf asymmetry, which tests for tolerance levels (see Table 1 for a complete description of hypotheses, predictions, and approaches).

**Table 1 plants-14-00049-t001:** Hypotheses, predictions, and approaches to evaluate leaf defense shifts in *Eriotheca gracilipes*.

Overview	Prediction	Approach	Resource
*H1. There is a turnover of defenses throughout leaf development*			
Survey 1—Physical defense	Physical defense peaks when leaves are fully expanded	Evaluation of leaf toughness	Figure 1A
Survey 2—Indirect defense (EFNs)	Indirect defense peaks on new flushed leaves	Evaluation of EFN activity	Figure 1B
*H2. Ants act as an indirect defense*			
Experiment 1	A: Ants decrease total herbivory rate	Evaluation of herbivory rate with and without ants	Figure 2
	B: Ants reduce herbivory on earlier foliar stages	Evaluation of herbivory rate throughout leaf development	Figure 2
*H3. Simulated herbivory increases ant visitation, decreasing herbivore numbers*			
Experiment 2	Different intensities of foliar simulated herbivory have different impacts on ant abundance	Assess ant attendance after foliar simulated herbivory	Figure 3A
Experiment 3	Abundance of herbivores vary over time after simulated herbivory	Observation of herbivores along time after simulated herbivory	Figure 3B
*H4. Different damage intensities impact differently the asymmetry of new flushed leaves*			
Experiment 4	High intensities of simulated herbivory affect asymmetry of new flushing leaves	Measurement of new flushing leaf asymmetry after simulated herbivory	Figure 4

**Figure 1 plants-14-00049-f001:**
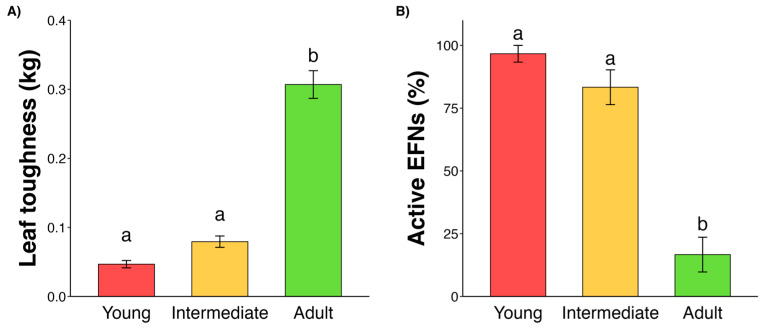
Leaf toughness increases during leaf development (**A**). Extrafloral nectary functionality decreases as leaves age and develop (**B**). Bars represent mean and standard error; letters represent statistical differences among groups by estimated marginal means (EMMs).

**Figure 2 plants-14-00049-f002:**
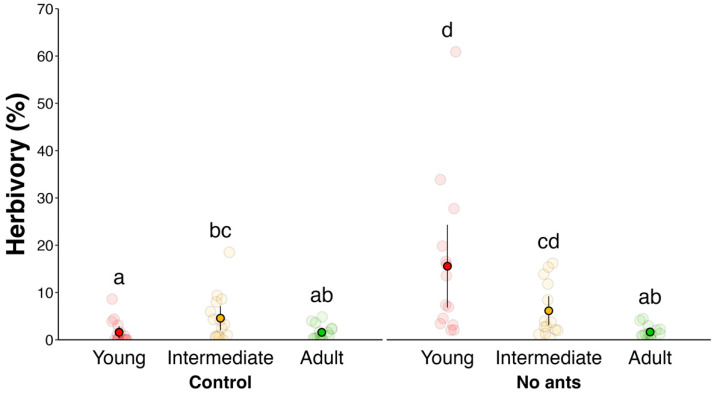
Herbivory damage (%) in *Eriotheca gracilipes* is higher in plants without ants compared to the control and highest in the young leaf stage (young = red; intermediate = dark yellow; adult = green). Letters indicate statistical differences among treatments based on estimated marginal means (EMMs). Bright solid circles represent the mean, vertical lines denote the standard error, and faint circles correspond to raw data points (individual leaves).

**Figure 3 plants-14-00049-f003:**
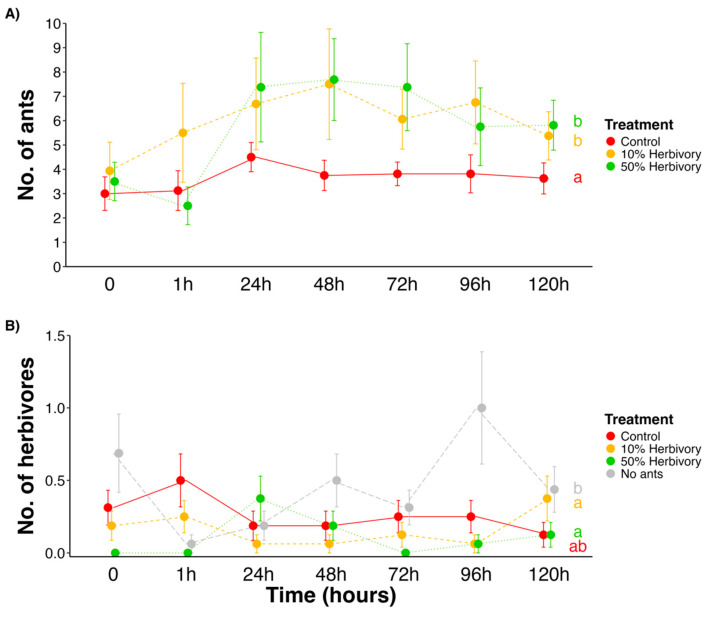
The number of ants is higher in the simulated herbivory treatments (10% and 50%), and it increases over time following the herbivory simulation (mean ± SE) (**A**). The number of herbivores is highest in the “No ants” treatment (**B**). Colors represent treatments (red = control; yellow = 10% herbivory; green = 50% herbivory; and gray = no ants), circles indicate the mean, and error bars represent the standard error. Letters denote statistically significant differences among treatments, based on estimated marginal means (EMMs) for the total number of ants (**A**) and herbivores (**B**).

**Figure 4 plants-14-00049-f004:**
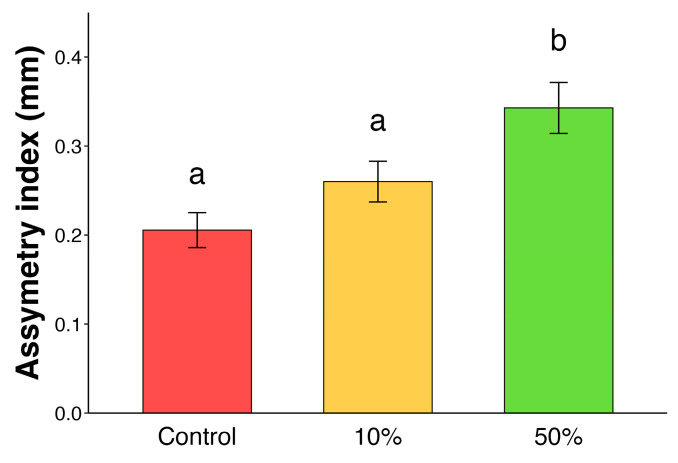
Variation in foliar asymmetry index in *Eriotheca gracilipes* between groups (control, 10%, and 50% of simulated herbivory). Bars represent the mean and standard error; letters represent significant differences among treatments by estimated marginal means (EMMs).

## 2. Results

**H1.** *There is a turnover of defenses throughout leaf development*.

Among treatments, the adult stage exhibited significantly higher leaf toughness compared to the young and intermediate stages (GLMM: χ^2^ = 297.2, df = 2, *p* < 0.001, Figure 1A). On average, adult leaves (0.307 ± 0.096 kg; mean ± standard deviation) were three times tougher than intermediate leaves (0.079 ± 0.039 kg) and six times tougher than young leaves (0.046 ± 0.025 kg). The number of active EFNs differed significantly across the leaf ontogenetic trajectory (GLMM: χ² = 31.4, df = 2, *p* < 0.001, Figure 1B). Young and intermediate leaves accounted for 97% and 83% of active EFNs, respectively, while adult leaves showed a marked decrease, with only 17% active EFNs.

**H2.** *Ants act as an indirect defense*.

Herbivory levels varied significantly based on the presence or absence of ants and leaf stage (young, intermediate, and adult) (GLMM: χ² = 26.2, df = 2, *p* < 0.001, Figure 2). Leaves in the “no ants” treatment received three times more damage compared to the “control” (GLMM: χ² = 15.1, df = 1, *p* < 0.001, Figure 2). Additionally, young leaves experienced one-and-a-half times more damage than intermediate leaves and five times more damage than adult leaves (GLMM: χ² = 18.5, df = 2, *p* = 0.001, Figure 2).

**H3.** *Simulated herbivory increases ant visitation, decreasing herbivore numbers*.

*Eriotheca* was visited by a total of ten ant species, in which *Camponotus crassus*, *Tapinoma* sp., and *Camponotus blandus* were the most abundant (Appendix A). A total of 20 herbivore morphospecies (Appendix A) were recorded, in which the most abundant were representatives of the families Chrysomelidae, Curculionidae, both Coleoptera, Oecophoridae, and Lepidoptera.

The number of ants differed significantly between treatments (control, 10%, and 50%) (GLMM: χ² = 7.4, df = 2, *p* = 0.02, Figure 3A), over time (GLMM: χ² = 65, df = 6, *p* < 0.001, Figure 3A), and in the interaction between treatments and time (GLMM: χ² = 27.6, df = 12, *p* = 0.006, Figure 3A). On average, both the 10% and 50% herbivory groups showed an increase in ant abundance of 1.6 and 1.5, respectively, compared to the control. Herbivore abundance also differed significantly among treatments (GLMM: χ² = 24.3, df = 3, *p* < 0.001, Figure 3B), with the “No ants” treatment having on average 4.5, 2.8, and 1.8 more herbivores than the “50% herbivory”, “10% herbivory”, and “Control” treatments, respectively. However, the number of herbivores did not vary significantly over time (GLMM: χ² = 3.9, df = 6, *p* = 0.6, Figure 3B).

**H4.** *Different damage intensities impact differently the asymmetry of new flushed leaves*.

Experimental groups had a significant difference in the asymmetry index after simulated herbivory (LM: F_(2,42)_ = 8.3, *p* < 0.001; Figure 4). The group of 50% of simulated herbivory presented the highest mean asymmetry index (0.34 ± 0.11 mm; mean ± standard deviation), differing significantly from the other two groups. On the other hand, the control and 10% of simulated herbivory groups showed similar results between them, in which the former presented an index of 0.20 ± 0.07 mm and the latter 0.26 ± 0.08 mm, showing no significant difference.

## 3. Discussion

### 3.1. Overview

In this study, we observed that *E. gracilipes* leaves undergo a defense turnover and EFNs exhibit inducibility after simulated herbivory. Additionally, *E. gracilipes* was tolerant to at least 10% of foliar removal, supporting our hypotheses. Specifically, we observed that (i) active EFNs peaked at young leaf stage; (ii) toughness peaked in adult leaves; (iii) ants reduced leaf herbivory in young leaves; (iv) simulated herbivory was responsible for an increase in ant frequency followed by a decrease in herbivore numbers; and (v) newly flushed leaves exhibited asymmetry after 50% of simulated herbivory but not after 10%. Based on this, *E. gracilipes* displays different defensive mechanisms throughout leaf development, before and after herbivore attack. Before damage, there is a defense turnover between the indirect defense mediated by EFNs and the physical defense represented by foliar toughness. After damage, defenses can be induced, and plants can tolerate at least 10% of foliar damage with no changes in foliar asymmetry in newly flushed leaves. Below, we discuss these results separately.

### 3.2. Before Damage—Turnover in Defense Strategies

**H1.** *There is a turnover of defenses throughout leaf development*.

*Eriotheca gracilipes* presented a turnover of defenses throughout leaf development. EFNs are highly active in young and intermediate leaves, and leaf toughness peaked in adult leaves. Several factors, such as resource availability and biotic pressures can drive defense turnover in plants [2,23]. Herbivores are one of the most important drivers. Essentially, herbivores can consume tissues from all foliar stages; however, they have preferences according to diet breadth, insect size, and feeding guild [25,26]. For instance, young leaves are heavily targeted by small herbivores that have piercing–sucking mouthparts because of its softness and nutritious tissues [25,26]. Moreover, as many herbivores are nitrogen deprived, they target young leaf stages because of its higher concentrations in nitrogen and water compared to fully expanded leaves [27]. Besides being highly targeted by herbivores, young leaves are important organs for plant development, and they are strong sink organs, demanding high resource allocation. Developmental constraints are also a strong driver in defense shifts throughout leaf development [28]. Leaf toughness is a natural process that occurs through lignification that hardens the leaf tissues and prevents expansion. And because unexpanded leaves need to develop and grow, leaf toughness is incompatible with young stages; as a result, leaf toughness is usually expressed in fully expanded leaves [28]. On the other side, defenses such as EFNs and chemical defenses are expressed in virtually all leaf stages; however, plants tend to concentrate defenses in young leaves [23,27,28]. Therefore, external and internal factors help to explain the turnover between defenses during foliar development.

**H2.** *Ants act as an indirect defense*.

Plants from the treatment group without ants had on average a higher rate of herbivory compared to the control group and young leaves had higher herbivore damage compared to other leaf stages. Ants are known as an effective indirect defense against plant natural enemies, and they are considered the main indirect defense in EFN-bearing plants [29,30,31]. In our study, we showed that EFNs are highly active in younger leaves, which explain the lower herbivory damage on young leaves from the control group. In the treatment group, young leaves had the highest herbivory rate among all leaf stages. Younger leaves are heavily targeted by herbivores for several factors, such as higher nutrient content, lower physical defenses, and synchrony with herbivores’ life cycle [28,32]. Therefore, *E. gracilipes* employed EFNs on young leaves to avoid the heavy pressure exerted by herbivores, especially by specialists, which is the case with the larvae from the family *Cerconota* sp. that feeds on young leaves (Appendix A).

### 3.3. After damage—Induced indirect defense and tolerance

**H3.** *Simulated herbivory increases ant visitation, decreasing herbivore numbers*.

Plants with simulated herbivory attracted a greater number of ants compared to the control group. Herbivory often induces an increase in nectar secretion from EFNs, which serves to recruit more ants to the plant. This enhanced nectar production can be an adaptive response to attract a higher number of protective ants after damage is detected, effectively deterring further herbivory [33,34]. However, that is only one of the possibilities; herbivory can also trigger the release of volatile organic compounds (VOCs) that indirectly signal ants to visit the plant. Some plants respond to herbivory by emitting VOCs that act as a “distress signal”, attracting more ants to protect the plant [35]. By recruiting ants preemptively, plants minimize the likelihood of subsequent attacks. This strategy, where plants simulate the presence of herbivores to boost ant visitation, is advantageous in environments with high herbivory pressure [36].

**H4.** 
*Different damage intensities impact differently the asymmetry of new flushed leaves.*


After simulated herbivory, newly flushing leaves of plants that suffered 50% simulated herbivory showed a more pronounced asymmetry compared to plants that suffered 10% simulated herbivory and control groups, which did not differ between each other. *Eriotheca gracilipes* showed no significant difference in leaf asymmetry between the 10% treatment and control group. Our results suggest that *E. gracilipes* might tolerate at least 10% defoliation. Past research has shown a relationship between leaf asymmetry and plant fitness [37,38]. In this case, there is a negative correlation between leaf asymmetry and plant fitness components [37,39,40]. Thus, we could expect that the fitness of the treatment with 10% simulated herbivory will not exhibit differences when compared to plants in the control group. Nonetheless, different herbivory intensities result in different outcomes based on plant species. For example, Ref. [41] showed that although plants possess tolerance to herbivore damage, each species responds differently. For instance, 10% leaf damage in *Piper arieianum* (Piperaceae) was sufficient to reduce plant fitness [42], while 25% leaf damage in *Raphanus raphanistrum* (Brassicaceae) did not affect reproductive output [41]. Leaf asymmetry is one of the parameters for developmental instability after stress [25]. Other studies have also reported leaf asymmetry because of herbivory [43]. Although herbivory is a common trophic process, high intensities of leaf loss can affect plant growth, survival, and primary production directly and indirectly. Therefore, plants’ capabilities to tolerate stress decreases as the environmental stress increases, resulting in higher leaf asymmetry [41,44].

### 3.4. Conclusions

In this study, we observed a dynamic interplay of defense mechanisms in *E. gracilipes* leaves, highlighting how plants adjust their strategies in response to herbivory and across different stages of leaf development. The findings support our hypotheses, demonstrating that *E. gracilipes* employs distinct defensive approaches that align with its developmental and ecological processes. Key observations include the following: (i) the peak activity of EFNs in young leaves; (ii) enhanced leaf toughness in adult leaves; (iii) reduced herbivory due to ant presence on young leaves; (iv) increased ant visitation and decreased herbivore presence following simulated herbivory; and (v) leaf asymmetry at higher herbivory levels.

These results emphasize that *E. gracilipes* strategically uses both indirect (ant recruitment via EFNs) and physical defenses (leaf toughness) across its leaf life cycle. Prior to herbivory, defense turnover occurs, with young leaves relying on active EFNs to deter herbivores attracted by higher nutrient content, while adult leaves adopt physical toughness as an effective defense. Following herbivory, the plant’s induced response leads to increased ant attraction, suggesting an adaptive mechanism to prevent further herbivore attack. Additionally, *E. gracilipes* shows tolerance to moderate herbivory (10% leaf removal) without significant developmental instability, as indicated by minimal leaf asymmetry, although higher herbivory (50%) causes greater asymmetry.

These observations underscore the importance of inducible defenses and herbivory tolerance, contributing to *E. gracilipes’* resilience in herbivore-rich environments. Overall, this study sheds light on the intricate balance between plant defenses and tolerance strategies, providing insights into the adaptive significance of EFNs and herbivory tolerance in maintaining *E. gracilipes* fitness across varying levels of herbivory stress.

## 4. Materials and Methods

### 4.1. Study Site and Plant Species

Fieldwork was conducted at the Ecological Reserve of Clube de Caça e Pesca Itororó in Uberlândia (48°17′27″ W; 18°58′30″ S), Uberlândia, Brazil. The vegetation is classified as Cerrado, predominantly characterized as shrubs and trees reaching heights of 2–8 m. The experiment was conducted in the border of this Cerrado area, where grass predominated alongside scattered herbs [45]. The climate is markedly seasonal with a drought from May to September and a rainy season from October to April, and an annual average precipitation of 1550 mm and temperature of 22 °C [46,47].

The selected plant species was *Eriotheca gracilipes* (Malvaceae), which is a common semi-deciduous tree in the Cerrado [30] and abundant in the study area. Its leaves are compound (usually with five leaflets), alternate, and glabrous [48]. Sapling plants have leaves going through three marked stages, namely newly flushed leaves (hereafter, “young”), not fully expanded leaves (hereafter, “intermediate”), and fully expanded leaves (hereafter, “adult”). These stages were separated based on their coloration. Young leaves have a reddish coloration (up to two weeks of age), the intermediates are brownish (two to four weeks of age), and the adult leaves are green (at least a month of age) (Appendix A). *E. gracilipes* has paired lateral EFNs at the base of the petiole and over the veins on the abaxial surfaces of leaves [49,50] which are visited by different arthropods (Appendix A), such as ants, spiders [10], and wasps [7].

We used only sapling plants, ranging from 70 cm up to 1.5 m in height and at least 10 m apart from each other. Plants had 5–10 branches, were under the same climatic conditions, and presented leaves in all three stages of development.

### 4.2. Experiments and Surveys

**H1.** *There is a turnover of defenses throughout leaf development*.

Survey 1—physical defense: To test for a peak of expression of the physical defense when leaves become adults, we evaluated leaf toughness in three leaf stages (young, intermediate, and adult). A total of 23 leaves were collected from each foliar stage (n = 69) from 23 plant individuals and stored in sealed bags inside a Styrofoam box with ice to prevent desiccation. Leaves were perforated twice in the central leaflet (middle leaflet out of five) at each side (left and right) beside the main vein, avoiding the main and secondary veins. We used a penetrometer (Fruit Hardness Tester MOD PTR-300) that measures in kilograms the force required to pierce the leaf. We fitted a generalized linear mixed-effect model (GLMM) to compare the foliar toughness in response to leaf stage using Gaussian distribution, in which the average of the two perforated holes was used as a response variable and plant individuals were considered as a random factor. Pairwise comparison was made using estimated marginal means.

Survey 2—indirect defense: To evaluate EFN activity intensity, leaves were selected in ten different individuals, with three leaves per plant (n= 90) and one at each foliar stage (young, intermediate, adult). We classified EFNs as “active” or “non-active” based on the necrosis evidence or lack thereof in the secretory tissues. The EFNs were considered non-active if they presented a dry appearance, black spots, and a total absence of extrafloral nectar and visiting ants. To compare the EFN activity among leaf stages, we fitted a GLMM with binomial distribution. The activity of EFNs was considered as the response variable and the plant individual was fitted as the random factor. Pairwise comparison was made using estimated marginal means.

**H2.** *Ants act as an indirect defense*.

Experiment 1: To evaluate if ants act as an indirect defense, 30 plants were selected and divided into two groups of 15 individuals each. The first group, “Treatment”, had all the ants and other arthropods manually removed and a non-toxic resin (Tanglefoot^®^ Rapids, Michigan, USA) applied upon the trunk at least 10 cm above ground. To prevent ants and other arthropods such as spiders from climbing the plants, all the nearby plant branches were removed to avoid “bridges” of contact. The second group, “Control”, received the resin only in half the circumference of the trunk, allowing free access of arthropods to foliage. In both groups, three leaves per individual were selected to measure herbivory. Photos were taken in the beginning and at the end of each stage to measure the levels of herbivory over leaf ontogeny. Foliar area loss was measured using the software ImageJ 1.47. To compare if foliar damage changed according to the presence of ants and foliar stage, we used a GLMM with Gaussian distribution. We used the interaction between the leaf stage and treatment as a fixed effect, and the leaf stage nested within plant individuals was used as random effect to control temporal repeated measures. Pairwise comparisons were performed using estimated marginal means.

**H3.** *Simulated herbivory increases ant visitation, decreasing herbivore numbers*.

Experiment 2: To verify if *E. gracilipes* shows evidence of inducible indirect defense, a simulated herbivory experiment was performed. For this, 60 plants not previously selected were tagged and divided into four groups of 15 individuals each. The group “No-Ants” had plants blocked from climbing arthropods with resin at 10 cm above ground, analogous to Experiment 1. In a second group, “Ants”, no type of manipulation was performed, only a layer of resin was put halfway around the main stem without blocking the passage of ants. In a third group, “10%”, we simulated herbivory by cutting 10% of the apical area of all leaves off the plant with scissors. In the last group, 50%, the same procedure was performed as in the 10% group, but 50% of the leaf area was removed. Plants from the 10% and 50% treatments also received Tanglefoot halfway around the main stem. The simulated herbivory was performed at 08:00 am. Ants and herbivores were observed in all plants at moment “0” (right before simulated herbivory) and at 1, 24, 48, 72, 96, and 120 h after simulated herbivory. All plants of all groups were observed for five minutes each. To compare ant and herbivore abundance between the groups analyzed after simulated herbivory, the total number of ants and herbivores per plant was used considering all observational periods (1, 24, 48, 72, 90, and 120 h) except time “0”. Then, we used a GLMM with Poisson distribution. Plant individuals were fitted as random effects. Pairwise comparisons were performed using estimated marginal means.

Survey 3: We observed the abundance of ants (except in the “No-Ants” group) and herbivores in all groups before (time “0”) and after simulated herbivory (1, 24, 48, 72, 90, and 120 h) to evaluate the effects of simulated herbivory on the associated ant fauna and their effect on herbivores. To compare if there is variation in the abundance of ants and herbivores per plant of each group between the analyzed schedules, we used a GLMM to evaluate ants’ abundance and a zero-inflated generalized linear mixed model (ZIGLMM) for herbivores, both with Poisson distribution. The time of evaluation nested within plant individuals was used as random effect to control temporal repeated measures in these two analyses (ant and herbivore abundance throughout time). A pairwise comparison was performed using Tukey’s HSD test for ants and estimated marginal means for herbivores when significant.

**H4.** *Different damage intensities impact differently the asymmetry of new flushed leaves*.

Experiment 3: To test tolerance, we used different intensities of simulated herbivory to assess the impacts on leaf asymmetry. This metric is commonly used as a biomarker of stress [51,52]. We did not evaluate reproductive components, because plants were still saplings and then did not flower. We selected three young leaves per plant (N = 9 leaflets per plant) that sprouted after the simulated herbivory (Experiment 2) in the ants, 10%, and 50% groups. After reaching the adult stage (~1 month), leaves were collected and photographed for analysis. We measured the distance from the margin to the midrib (horizontally) on both left and right sides of the widest leaflet region. With these measurements, we averaged the absolute value of the subtraction of the measures between the sides for each plant, resulting in a leaf asymmetry index. For this analysis, we used a linear model in which we used asymmetry. Pairwise comparison was performed using Tukey’s HSD test.

## Data Availability

The original data presented in the study will openly be available in “Dryad” after formal publication.

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
