# Peer review of "Sequential Defense Strategies: From Ant Recruitment to Leaf Toughness"

_plants, 2024, doi:10.3390/plants14010049_

Round 1
Reviewer 1 Report
Comments and Suggestions for Authors
Reviewer Comments
Title: Plant defenses turn over and response after simulated herbivory
Authors: Danilo F. B. dos Santos , Eduardo S. Calixto , Helena M. Torezan-Silingardi , Kleber Del-Claro
This study explored the adaptive defense strategies of Eriotheca gracilipes, focusing on extrafloral nectaries (EFNs) and the tolerance of herbivory across the stages of leaves from young to intermediate to adult developmental stages. Well done.
Introduction: The review of literature and the test hypotheses were sufficiently comprehensive and targeted to the study goals. See line 69 for a typo: scaping should be escaping.
Methods and Materials were described in sufficient detail to be replicated by others.
Results: Findings were appropriately analyzed with nicely delineated bar graphs. Two minor and easy to fix issues are as follows:
Figure 1b: The dependent variable is denoted as decimals (proportions) and not percents.
Figure 2: I am curious as to why Figure 2 differed in style relative to Figure 1 and 4.
Conclusion: Eriotheca displayed different defensive mechanisms throughout leaf development, before and after herbivore a􀄴ack and that this plant can tolerate at least 10% of foliar damage with no changes in foliar asymmetry in newly flushed leaves was an important finding. Overall, this study provides insight into the adaptive significance of EFNs and herbivory tolerance in across varying levels of herbivory stress in Eriotheca.
Author Response
We sincerely appreciate the reviewers' thoughtful and constructive comments on our manuscript. The feedback has provided valuable insights and helped us strengthen the clarity and rigor of our work. We are grateful for the time and effort they invested in thoroughly reviewing our study and for their suggestions, which have significantly improved the quality of our research. Thank you for your careful consideration and expertise.
Reviewer 1:
This study explored the adaptive defense strategies of Eriotheca gracilipes, focusing on extrafloral nectaries (EFNs) and the tolerance of herbivory across the stages of leaves from young to intermediate to adult developmental stages. Well done.
Introduction: The review of literature and the test hypotheses were sufficiently comprehensive and targeted to the study goals. See line 69 for a typo: scaping should be escaping.
#1: Thank you, we have now fixed (see line 69).
Methods and Materials were described in sufficient detail to be replicated by others.
#2: Thank you!
Results: Findings were appropriately analyzed with nicely delineated bar graphs. Two minor and easy to fix issues are as follows:
Figure 1b: The dependent variable is denoted as decimals (proportions) and not percents.
#3: Thank you! We have fixed figure 1b (see Figure 1).
Figure 2: I am curious as to why Figure 2 differed in style relative to Figure 1 and 4.
#4: I have used figure 4 in a different style to provide a more visual descriptive information of the data. Since this figure represents one of the most important data in the paper. However, if the reviewer disagrees, I can upload a bar version of the data.
Conclusion: Eriotheca displayed different defensive mechanisms throughout leaf development, before and after herbivore a?ack and that this plant can tolerate at least 10% of foliar damage with no changes in foliar asymmetry in newly flushed leaves was an important finding. Overall, this study provides insight into the adaptive significance of EFNs and herbivory tolerance in across varying levels of herbivory stress in Eriotheca.

Reviewer 2 Report
Comments and Suggestions for Authors
In this MS, the authors carried out a series of experiments to study the dynamic defense strategies of Eriotheca gracilipes, focusing on extrafloral nectaries (EFNs) and tolerance across leaf development stages. The topic is very interesting, while there are many shortcomings which are following as:
Q1: Title: Confused title that not clear for plant defenses or simulated herbivory! And the topic of this study is that ants can act as an indirect defense due to EFN presence. While no any information about ants or EFN.
Q2: Abstract: L14: “these aspects are neglected in plant-herbivore studies in tropical systems" Just plant-herbivore studies not considering that ants can act as an indirect defense due to EFN presence? Just plant-herbivore studies in tropical systems not other systems were neglected? Confused! L16+17: “from indirect defenses (EFNs) in young leaves” and “. Simulated herbivory” Is EFNs indirect defense? EFNs is indirect defense due to ants visiting. How to simulate herbivory? Simplify to introduce the simulated herbivory, here.
Q3: Table 1: Not Table 1 and the table title were found in this MS. And this table should be removed to M&M. And for H2, A (Ants decrease herbivory rate) and B (Ants reduce herbivory on earlier foliar stages) were same meaning just on earlier foliar stages for B not for A.
Q4: Figure 1: Mature leaves not adult leaf. The values signed as “a” were lower than that signed as “b” in Fig.1 A, while the values signed as “a” were higher than that signed as “b” in Fig.1B. Not inconsistent for the signed as a or b. And this is easy to say that leaf toughness increases during leaf development, a selfevident process. And here, the unit of leaf toughness (kg) is not clear, per leaf or per cm2 of leaf? And what about Active EFNs (%)? Not clear!
Q5: Figure 2: What about herbivors? Not just to say herbivory damage. For the herbivor aphids, as ant existing to protect them, the abundance of aphid population may be increased not reduced. And how about the density of the control with ants? Not given.
Q6: Figure 3: For the Control (no herbivory), there were ants in Fig.3A. While for the control in Fig.3B it is also no herbivory, why there were herbivory for the control in Fig.3B? The control should be “no ants” in Fig.3B.
Q7: Figure 4: What about the used developmental leaf? Young, intermediate or mature leaf? Or the three developmental stages of leaves should be studied here.
Q8: 4.2. Experiments and Surveys: H1: “We classified EFNs as “active” or “non-active” based on the necrosis evidence or lack thereof in the secretory tissues [24]” Give pictures to show the necrosis evidence (active) or lack thereof (non-active) in the secretory tissues, not just said that cited the [24] reference. H4: Provided the photos to show the asymmetry of treated leaves.
Other comments were directly marked in the PDF file.

Author Response
We sincerely appreciate the reviewers' thoughtful and constructive comments on our manuscript. The feedback has provided valuable insights and helped us strengthen the clarity and rigor of our work. We are grateful for the time and effort they invested in thoroughly reviewing our study and for their suggestions, which have significantly improved the quality of our research. Thank you for your careful consideration and expertise.
Reviewer 2:
In this MS, the authors carried out a series of experiments to study the dynamic defense strategies of Eriotheca gracilipes, focusing on extrafloral nectaries (EFNs) and tolerance across leaf development stages. The topic is very interesting, while there are many shortcomings which are following as:
Q1: Title: Confused title that not clear for plant defenses or simulated herbivory! And the topic of this study is that ants can act as an indirect defense due to EFN presence. While no any information about ants or EFN.
#1: Thank you for the comment! I have now updated the title to make it clearer and explicit. (L2-3): Sequential defense strategies: from ant recruitment to leaf toughness.
Q2: Abstract: L14: “these aspects are neglected in plant-herbivore studies in tropical systems" Just plant-herbivore studies not considering that ants can act as an indirect defense due to EFN presence? Just plant-herbivore studies in tropical systems not other systems were neglected? Confused! L16+17: “from indirect defenses (EFNs) in young leaves” and “. Simulated herbivory” Is EFNs indirect defense? EFNs is indirect defense due to ants visiting. How to simulate herbivory? Simplify to introduce the simulated herbivory, here.
#2: Thank you for the insights! I addressed each one of them as follow:
- I have elaborated on the scientific gap in the abstract. Now it reads as: “However, leaf defense shifts during leaf development, including extrafloral nectaries (EFNs), are neglected in in natural tropical systems” (L13-15).
- EFNs are not itself indirect defenses, they are glands produced by the plants. To improve the clarity of the sentence I have included “e.g.” to make it clear that EFNs can function as indirect defense by attracting ants. Now, it reads as: “shifting from indirect defenses (e.g. EFNs) in young leaves to physical defenses in adult leaves” (L16-17).
- I have now introduced how simulated herbivory was made in the paper, the sentence reads as: We also simulate herbivory by cutting the leaves to address the role of visiting ants against herbivores” (L17-18).
Q3: Table 1: Not Table 1 and the table title were found in this MS. And this table should be removed to M&M. And for H2, A (Ants decrease herbivory rate) and B (Ants reduce herbivory on earlier foliar stages) were same meaning just on earlier foliar stages for B not for A.
#3: Thank you for the comments! Indeed, I had forgotten to add the table title. The table title reads as: “able 1: Hypotheses, predictions and approaches to evaluate leaf defense shifts in Eriotheca gracilipes” (L86).
Regarding your second comment, although they do seem similar, they mean two different approaches. The first, ants reduce herbivory, concerns to the treatments in general, which means that ants will protect leaves regardless of age. The second prediction concerns to the leaf stages, since younger leaves are more susceptible to herbivory, we predicted that ants would reduce herbivory in younger leaves. I hope that this has clarified your point.
Q4: Figure 1: Mature leaves not adult leaf. The values signed as “a” were lower than that signed as “b” in Fig.1 A, while the values signed as “a” were higher than that signed as “b” in Fig.1B. Not inconsistent for the signed as a or b. And this is easy to say that leaf toughness increases during leaf development, a selfevident process. And here, the unit of leaf toughness (kg) is not clear, per leaf or per cm2 of leaf? And what about Active EFNs (%)? Not clear!
#4: Thank you for noticing the interchangeable use of “adults” and “mature”. I have standardized the paper using only the “adult” term to refer to leaves fully developed.
Regarding your concern about the letters “a” and “b” in Figure 1, these letters represent the results of a post hoc test, indicating which treatments differ from others. They are used solely to distinguish what is statistically different and what is not. Therefore, the choice of “a” or “b” is merely symbolic of statistical significance. Additionally, when using letters to differentiate treatments, we follow alphabetical order, meaning that “a” will always appear first, regardless of the magnitude of the treatment it represents. I hope this explanation addresses your concern.
The unit of leaf toughness is described in the methods (L289-291). Basically, we used this device to pierce through the leaf, and the strength used by the piercing point of the device to perforate the leaf was measured in Kg. I hope this have clarified your point.
Q5: Figure 2: What about herbivors? Not just to say herbivory damage. For the herbivor aphids, as ant existing to protect them, the abundance of aphid population may be increased not reduced. And how about the density of the control with ants? Not given.
#5: Thank you for the comment. First, I think it is important to clarify that Figure 2 concerns to the herbivory damage, in this case, translated in leaf are loss (percentage of leaf loss). Therefore, even knowing that ants tend aphids and can increase their population, the figure does not concern about herbivore population. Additionally, in our study we aimed to understand how ants reduce leaf herbivory. Leaf area is a measurable trait that can be quantified and translated into information. Based on this, we did not try to quantify piercing/sucking herbivores, since they offer the challenge of quantifying the herbivory caused. We do recognize the importance and effects of aphids as herbivores, however, for the scope of the study, we focused on herbivores that could visibly remove leaf area.
Regarding your comment about the density of the control with ants, I’m afraid to say that I did not understand the point made.
Q6: Figure 3: For the Control (no herbivory), there were ants in Fig.3A. While for the control in Fig.3B it is also no herbivory, why there were herbivory for the control in Fig.3B? The control should be “no ants” in Fig.3B.
#6: I believe the reviewer may have misunderstood the figure. Figure 3 illustrates the abundance of ants and herbivores across different treatments. We measured the number of ants on control plants, plants subjected to 10% simulated herbivory, and plants subjected to 50% simulated herbivory. For the number of herbivores, we also included the treatment "no-ants."
With this figure, we aimed to understand how many herbivores visit plants that have ants but no simulated herbivory (control), plants with 10% herbivory, plants with 50% herbivory, and plants with no simulated herbivory and no ants. I kindly suggest that the reviewer refer to the Methods section (L322–330), where we describe the four treatments used in the experiment. I hope this have clarified your point.
Q7: Figure 4: What about the used developmental leaf? Young, intermediate or mature leaf? Or the three developmental stages of leaves should be studied here.
#7: Our prediction was that plants with simulated herbivory would not produce new leaves with balanced asymmetry. Which means that the asymmetry should be measured in leaves from damaged plants. In this case, we did not separated leaf in developmental stages because we were not concerned about the asymmetry in developmental stage, but in the intensity of herbivory caused to the plant. I hope this have clarified your point.
Q8: 4.2. Experiments and Surveys: H1: “We classified EFNs as “active” or “non-active” based on the necrosis evidence or lack thereof in the secretory tissues [24]” Give pictures to show the necrosis evidence (active) or lack thereof (non-active) in the secretory tissues, not just said that cited the [24] reference. H4: Provided the photos to show the asymmetry of treated leaves.
#8: Unfortunately, we did not take photographs of the nectaries. However, we believe the classification is straightforward. Regarding the asymmetry of the leaves, we do have a photographic record. However, it seems unnecessary to include these images unless for purely illustrative purposes, as the raw data will be made available in an open repository. I hope this have satisfied your point.

Round 2
Reviewer 2 Report
Comments and Suggestions for Authors
The version has been revised based on the comments of the reviewers. While there are some shortcomings which were following as:
Q1: Key words: Too general not specific for this study! Suggest to add some specific topics of this study, e.g., extrafloral nectaries, leaf toughness, ant visiting etc.
Q2: Not clear about herbivorous species and ant species, so the supplementary tables (Table S1 and Table S2) should be moved to the results. Not just as supplementary tables.
Q3: Figure 3: No significant differences were analyzed among differernt treatments, and no unit was noted .
Q4: Table S1 and Table S2 were revised. No clear unit (per leaf or per plant) was noted in these two tables, and no significant difference was analyzed among different treatments.
Q5: The abundance of ants was higher, while too low abundance or density (much more 0, 1 or 2 of herbivorous pests was given in Table S2, which couldn't be used to presume the general conclusion that "Sequential defense strategies: from ant recruitment to leaf toughness".

Author Response
We sincerely appreciate the reviewers' thoughtful and constructive comments on our manuscript. The feedback has provided valuable insights and helped us strengthen the clarity and rigor of our work. We are grateful for the time and effort they invested in thoroughly reviewing our study and for their suggestions, which have significantly improved the quality of our research. Thank you for your careful consideration and expertise.
Q1: Key words: Too general not specific for this study! Suggest to add some specific topics of this study, e.g., extrafloral nectaries, leaf toughness, ant visiting etc.
#1: Thank you for the suggestions. I have now included more specific key words.
Q2: Not clear about herbivorous species and ant species, so the supplementary tables (Table S1 and Table S2) should be moved to the results. Not just as supplementary tables.
#2: The table describing the ant and herbivore species are very descriptive and can be used to trace back the species identity if someone is curious enough to check. However, the main idea of the paper is to show how defenses change during leaf development. Based on that, I believe that the tables would be extra information that would not directly contribute to the idea.
Q3: Figure 3: No significant differences were analyzed among differernt treatments, and no unit was noted.
#3: I may have to disagree with the reviewer. The post-hoc test is shown in the figure (and described in the figure legend), different letters represent significant differences among treatments. And I’m not sure if I understand the “no unit was noted”. In X axis there is time in hours and Y axis is the number of ants/herbivores.
Q4: Table S1 and Table S2 were revised. No clear unit (per leaf or per plant) was noted in these two tables, and no significant difference was analyzed among different treatments.
#4: Thank you for the suggestion. I have now added the unit in the title of the table.
Q5: The abundance of ants was higher, while too low abundance or density (much more 0, 1 or 2 of herbivorous pests was given in Table S2, which couldn't be used to presume the general conclusion that "Sequential defense strategies: from ant recruitment to leaf toughness".
#5: The general conclusion is not based on the ant/herbivores descriptive table. The main conclusion was based on the defenses employed by the leaf during its development. I’m afraid the reviewer may have confused the information presented.

Round 3
Reviewer 2 Report
Comments and Suggestions for Authors
This version has been revised based on the comments of the reviewers.